# It Takes Two to Tango: Potential Prognostic Impact of Circulating TGF-Beta and PD-L1 in Pancreatic Cancer

**DOI:** 10.3390/life12070960

**Published:** 2022-06-26

**Authors:** Ingrid Garajová, Andrea Cavazzoni, Michela Verze, Roberta Minari, Giuseppe Pedrazzi, Rita Balsano, Fabio Gelsomino, Raffaele Dalla Valle, Graziana Digiacomo, Elisa Giovannetti, Francesco Leonardi

**Affiliations:** 1Medical Oncology Unit, University Hospital of Parma, 43126 Parma, Italy; ingegarajova@gmail.com (I.G.); mverze@ao.pr.it (M.V.); rominari@ao.pr.it (R.M.); rita.balsano@gmail.com (R.B.); fleonardi@ao.pr.it (F.L.); 2Department of Medicine and Surgery, University of Parma, 43126 Parma, Italy; andrea.cavazzoni@unipr.it (A.C.); giuseppe.pedrazzi@gmail.com (G.P.); raffaele.dallavalle@unipt.it (R.D.V.); 3Lab of Medical Oncology, Department of Medical Oncology, Cancer Center Amsterdam, Amsterdam UMC, VU University Medical Center (VUmc), 1081 HV Amsterdam, The Netherlands; 4Department of Oncology and Hematology, University Hospital of Modena, 41125 Modena, Italy; fabiogelsomino83@yahoo.it; 5Cancer Pharmacology Lab, AIRC Start-Up Unit, Fondazione Pisana per la Scienza, 56124 Pisa, Italy

**Keywords:** pancreatic adenocarcinoma, prognostic biomarker, liquid biopsy, sTGF-beta, sPD-L1

## Abstract

**Background:** Pancreatic ductal adenocarcinoma (PDAC) is a highly devastating disease with rising incidence and poor prognosis. The lack of reliable prognostic biomarkers hampers the individual evaluation of the survival and recurrence potential. **Methods:** Here, we investigate the value of plasma levels of two potential key players in molecular mechanisms underlying PDAC aggressiveness and immune evasion, soluble TGF-beta (sTGF-beta) and sPD-L1, in both metastatic and radically-resected PDAC. To this aim we prospectively enrolled 38 PDAC patients and performed appropriate statistical analyses in order to evaluate their correlation, and role in the prediction of disease relapse/progression, and patients’ outcome. **Results:** Metastatic patients showed lower levels of circulating sTGF-beta and higher levels of sPD-L1 compared to radically-resected patients. Moreover, a decrease in sTGF-beta levels (but not sPD-L1) was significantly associated with disease relapse in radically-resected patients. We also observed lower sTGF-beta at disease progression after first-line chemotherapy in metastatic patients, though this change was not statistically significant. We found a significant correlation between the levels of sTGF-beta and sPD-L1 before first-line chemotherapy. **Conclusions:** These findings support the possible interaction of TGF-beta and PD-L1 pathways and suggest that sTGF-beta and sPD-L1 might synergize and be new potential blood-based biomarkers.

## 1. Introduction

With a five-year survival rate of 10%, pancreatic ductal adenocarcinoma (PDAC) remains one of the most difficult tumors to treat [1,2]. Radical surgery is the only potentially curative treatment, though, even in the group of radically resected patients, the five-year survival rate is below 25% [3]. There are several reasons for the dismal prognosis of PDAC: the retroperitoneal location with late onset of symptoms, the biological aggressiveness characterized by early metastasis—around 50% of patients have metastatic disease at presentation [4,5], and the impressive resistance to many anticancer agents [6,7,8,9].

The current diagnostic methodologies for PDAC are based on clinical symptoms, radiological data, and pathological confirmation using surgical or biopsy specimens. Such specimens are also used for molecular characterization, that should improve clinical management [10]. However, fine-needle aspiration techniques cannot always be performed because of the PDAC retroperitoneal position [11]. In addition, these techniques are invasive, and have a modest sensitivity due to the high stromal density of PDAC [12,13].

There is thus a strong need for the use of more accessible materials implying non- or minimally invasive procedures which would allow systematic and real-time monitoring of cancer molecular alterations in patients, such as using liquid biopsy approaches [14,15]. Notably, clinical application of liquid biopsies in PDAC patients would include both better early/differential diagnosis and identification of prognostic or predictive biomarkers for disease relapse after radical surgery, or biomarkers for disease progression in metastatic settings to determine the potential chemosensitivity or chemoresistance and guide clinical management [16].

However, unlike other solid tumors, such as lung cancer, where liquid biopsy tests are currently applied to clinical practice [17,18], the only blood-derived biomarker currently used for PDAC is carbohydrate antigen 19-9 (CA19-9, also known as cancer antigen 19-9 or sialylated Lewis antigen), which is neither diagnostic nor specific for PDAC clinical monitoring [2,9,19]. Indeed, false negative results are found in approximately 5–10% of individuals, who are Lewis’s antigen negative because of the *Lewis (a-b)* genotype while high levels of CA19-9 are often found in both patients with PDAC and other periampullary cancers, as well as in patients with obstructive jaundice due to benign diseases [20]. Thus, CA19-9 is mainly used in monitoring conditions and follow-up after therapy [21]. However, reductions in the levels of CA 19-9 in response to therapy were not associated with survival and therefore do not provide a valid marker for survival when patients begin therapy [22].

Previous studies on new potential blood-based diagnostic/prognostic biomarkers have evaluated several proteins such as thrombospondin and AXL [23], which seem to play a role in PDAC carcinogenesis and progression. Indeed, two of the most important aspects of a reliable biomarker are: (1) the strength of its connection to the disease, and (2) its relationship to other disease-specific biomarkers [24].

The transforming growth factor-beta (TGF-beta) pathway has been identified as a major contributor to the pathogenesis of PDAC: the TGF-beta signaling pathway might act both as a tumor promoter and as a tumor suppressor according to the tumor stage and surrounding tumor microenvironment (TME) [25]. In advanced PDAC stages, TGF-beta contributes to cancer progression by induction of epithelial-mesenchymal transition (EMT), promoting tumor stemness potential and proliferation [26]. A recent study showed that pretreatment soluble TGF-beta (sTGF-beta) levels can serve as a prognostic indicator in unresectable pancreatic cancer patients treated with FOLFIRINOX chemotherapy [27]. Likewise, the dynamics of sTGF-beta during chemotherapy showed prognostic value. However, because of the small sample size this study could only confirm that patients who achieved partial response as the best response showed statistically increased sTGF-beta levels at the time of disease progression compared to the first response assessment, while significant differences could not be shown in other analyses, prompting further studies.

Of note, TGF-beta directly suppresses immune cells [28,29] and has a pivotal role in PDAC immune evasion. Indeed, loss of the TGF-beta target SMAD4, which is deleted in approximately 55% of PDACs, was associated with a poor T-cell infiltrate. Conversely, SMAD4 indirectly promotes programmed death 1 ligand 1 (PD-L1) expression in the PDAC TME by enhancing T-cell infiltration and IFNγ biosynthesis [30].

Most importantly, therapy targeting immune checkpoint expressed by activated T cells, such as programmed death 1 (PD1) or PD-L1, has been successfully developed, alone or in combination with chemotherapies, for the treatment of various tumors [31,32]. This therapeutic approach failed in PDAC. However, TGF-beta negatively regulates the expression of PD-L1 on the pancreatic neoplastic epithelium [29], and some studies suggest a correlation between TGF-beta and the soluble programmed death ligand 1 (sPD-L1), which is released from PD-L1-positive cells and plays an important role in immunoregulation. The interplay between PD-L1 and TGF-beta seems indeed essential to control fibroproliferation, and the stimulation of fibroblasts with TGF-beta increased the secretion of extracellular vesicles containing PD-L1 [33].

Thus, the addition of TGF-beta inhibitors in selected immunotherapy regimens might have synergistic effects. Furthermore, sPD-L1 participates in tumor-associated immune suppression and host immune damage leading to cancer progression [34]. These findings support further studies on liquid biopsies to further understand the impact of sPD-L1 on PDAC prognosis as well as its potential interaction with TGF-beta in order to stratify patients for future trials of immunotherapies combined with anti-TGF strategies in selected populations.

Therefore, in this study, we aimed to measure both sTGF-beta and sPD-L1 in the plasma of radically resected and metastatic PDAC patients in order to evaluate their clinical implication in the prediction of disease relapse/progression, as well as their correlation with patients’ outcome.

## 2. Materials and Methods

### 2.1. Study Population

The study design and protocol were approved by the local Medical Ethics Board of the University Hospital of Parma (Parma, Italy), in accordance with the ethical guidelines of the Declaration of Helsinki. Before study participation, written informed consent was obtained from all participants. Both metastatic and radically resected PDAC patients were prospectively enrolled in this study. Only patients (>18 years old) with cytologic or histologic confirmed diagnosis of PDAC were included. Clinicopathological characteristics were collected in a prospectively maintained database. The clinically relevant variables that were extracted included age, sex, ECOG PS, date of surgical resection for pancreatic cancer, histology/cytology, and TNM. We also collected data of different immune cells (total white blood cells, neutrophils, monocytes, and lymphocytes).

Moreover, date of relapse after surgery, date of progression after palliative chemotherapy, type of chemotherapy, and date of last follow-up or date of death was collected.

### 2.2. Detection of Soluble PD-L1 (sPD-L1) and Soluble TGF-Beta (sTGF-Beta) Levels

Soluble PD-L1 (sPD-L1) and soluble TGF-beta (sTGF-beta) levels were determined in the blood of PDAC patients at the Laboratory of the Department of Medicine and Surgery (Parma, Italy). For the radically resected patients the PB samples were collected at radical surgery and at the disease progression. For the metastatic patients the PB samples were collected before starting the first-line palliative treatment and at the disease progression. Plasma samples were collected and stored at −80 °C. Then, specific enzyme-linked immunosorbent assay (ELISA) assays (R&D System, Minneapolis, MN, USA) were employed to determine the amount of sPD-L1 (kit#DB7H10) or sTGF-beta (kit#DB100B) in each sample following the manufacturer’s instructions.

### 2.3. Statistical Analysis

Descriptive statistics were used to describe the patients’ characteristics and outcomes. Correlations between sPD-L1 and sTGF-beta levels and patients’ clinical outcome were provided using the Student’s *t*-test. Clinicopathologic and long-term survival data were collected and reviewed to explore the prognostic/predictive implication of sTGF-beta and sPD-L1. Overall survival (OS) in metastatic PDAC was calculated from the date of diagnosis of metastatic disease to the date of death or date of the last follow-up. Disease-free survival (DFS) was calculated from the date of pancreatic surgery to the date of disease recurrence. OS and DFS curves were constructed using Kaplan–Meier method, and differences were analyzed using log-rank (Mantel–Cox) test. A *p* value of 0.05 was set as a threshold of statistical significance. IBM SPSS Statistics v 25.0 (IBM) was used to perform all computational analyses.

## 3. Results

### 3.1. Patient Characteristics

The present study included a total of 38 PDAC patients, of which the baseline clinico- pathological characteristics are summarized in Table 1. These patients were prospectively enrolled from 2019 to 2020 at the Medical Oncology Unit of the University Hospital of Parma. Out of 38 PDAC patients included in our study, 23 (60.5%) were metastatic at the time of diagnosis. Median OS for metastatic group was 9 months (95% CI, 1–23 months), while median DFS for radically resected PDAC patients was 8 months (95% CI, 4–18 months). The mean age of the patients was 69 years, both for metastatic and radically resected PDAC patients. Moreover, 56.6% and 47% of PDAC patients were females in metastatic and radically resected cohort, respectively.

### 3.2. Circulating sTGF-Beta and sPD-L1 Levels in Radically-Resected and Metastatic PDAC Patients

To investigate the potential clinical value of sTGF-beta and sPD-L1 in the cohorts of patients with histologically confirmed PDAC, we successfully performed ELISA assays, showing that baseline levels varied between the groups. The distribution of sTGF-beta levels in the non-metastatic/radically resected group resulted in an average value of 15,968 ± (standard error, SE) 3530 pg/mL, whereas the distribution of sTGF-beta levels in the metastatic group resulted in an average value of 8730 ± (standard error, SE) 1609 pg/mL. Regarding PD-L1 levels, in the non-metastatic group we measured an average value of 94.17 ± (standard error, SE) 5.42 pg/mL, while in the metastatic group the resulting average value was 135.70 ± (standard error, SE) 12.59 pg/mL.

Statistical analyses showed a significant difference for TGF-beta values between the metastatic (n = 23) and radically resected PDAC patients (n = 15) (Student’s *t*-test, *p* = 0.042), with significantly lower levels of TGF-beta in metastatic patients in comparison to radically resected PDAC patients (Figure 1A).

Conversely, we observed a statistically significant difference for PD-L1 values between the metastatic and radically resected PDAC patients (Student’s *t*-test, *p* = 0.010). In particular, resected patients had significantly lower levels of PD-L1 compared to metastatic PDAC patients (Figure 1B).

### 3.3. Circulating TGF-Beta and PD-L1 Levels in Radically-Resected PDAC Patients at Disease Relapse

Considering the radically resected PDAC patients who experienced disease relapse (n = 10), the distribution of baseline sTGF-beta levels had an average value of 17,728 ± (standard error, SE) 3898 pg/mL, while at disease relapse the average sTGF-beta plasma value was 12,038 ± (standard error, SE) 2968 pg/mL (Figure 2A).

However, sPD-L1 levels in the non-metastatic group had similar average values immediately post-surgery and at disease relapse after radical surgery (95.04 ± 6.312 pg/mL vs. 93.91 ± 6.193 pg/mL (Figure 2B).

Thus, we observed a statistically significant difference for sTGF-beta values in radically resected PDAC patients after surgery in comparison to sTGF-beta values at disease relapse (Student’s *t*-test, *p* = 0.031); in particular, we measured less TGF-beta at disease relapse in comparison to basal values immediately after radical resection. The same analysis for PD-L1 levels did not show any statistically significant difference (Student’s *t*-test, *p* = 0.891).

The statistically significant difference for sTGF-beta values in radically resected PDAC patients after surgery in comparison to sTGF-beta values at disease relapse has been shown also in the correlation matrix reported in the Figure 3 (*p* = 0.049).

### 3.4. Circulating sTGF-Beta and sPD-L1 Baseline Levels in Metastatic PDAC Patients at Disease Progression after First-Line Chemotherapy

The average baseline levels of sTGF-beta in the metastatic group (n = 23) were 24,945 ± (standard error, SE) 5546 pg/mL, while at disease progression after first-line chemotherapy were slightly reduced to 20,682 ± 2586 pg/mL (Figure 4A).

Similarly, we observed only a minor modulation of the distribution of PD-L1 levels in the non-metastatic group results, with an average value of 77.60 ± (standard error, SE) 8.289 pg/mL at baseline, versus an average value of 83.99 ± (standard error, SE) 7.451 pg/mL at disease progression after first-line chemotherapy (Figure 4B).

Thus, we did not observe a statistically significant difference for sTGF-beta values nor for PD-L1 in this setting, though, in agreement with previous findings [27], we observed lower levels of TGF-beta at metastatic PDAC disease progression.

### 3.5. Circulating TGF-Beta and PD-L1 and Proportions of Different Immune Cells

Together with soluble TGF-beta and PD-L1, all patients’ blood samples were examined for different immune cells (total white blood cells, neutrophils, monocytes, and lymphocytes). In particular, we also evaluated the NLR (ratio neutrophils/lymphocytes) by investigating the correlation of metastatic PDAC patients’ OS with “high” NLR (higher than median) compared to “low” NLR (lower than median), with no significant difference between the two groups (*p* = 0.6). Similarly, the OS of resected PDAC patients with “high” NLR compared to “low” NLR did not show a significant difference (*p* = 0.058), as reported in the Appendix A.

### 3.6. Circulating TGF-Beta and PD-L1 and Clinical Outcome

#### 3.6.1. Metastatic PDAC and Baseline sTGF-Beta and sPD-L1

The median OS for the patients on the metastatic group was 9 months (95% CI, 6.0–11.9 months). Metastatic PDAC patients with baseline upregulation (i.e., values above the average) of sTGF-beta had median OS of 8 months (95% CI, 5.84–10.15 months), while metastatic PDAC with downregulated (i.e., values below the average) sTGF-beta had a median OS of 11 months (95% CI, 6.35–15.64 months). However, this difference was not statistically significant (*p* = 0.375, Figure 5A).

Similarly, metastatic PDAC patients with baseline upregulation of sPD-L1 had a median OS of 8 months (95% CI, 4.6–11.39 months), while metastatic PDAC with downregulated sTGF-beta had a median OS of 10 months (95% CI, 7.07–12.92 months, *p* = 0.218, Figure 5B).

#### 3.6.2. Radically Resected PDAC and Baseline sTGF-Beta and sPD-L1

The median DFS for the group of radically resected PDAC patients was 8 months (95% CI, 5.07–10.92 months). Patients with baseline upregulation of sTGF-beta had median DFS of 9 months (95% CI 1–17 months), in comparison to patients with downregulated sTGF-beta with median DFS of 7 months (95% CI 4.6–9.4 months). This difference was not statistically significant (*p* = 0.537, Figure 6A).

No significant differences were also observed when evaluating radically resected PDAC patients with baseline upregulation of sPD-L1, who had median DFS of 9 months (95% CI, 2.55–15.44 months), in comparison to resected PDAC with downregulated sPD-L1-beta, showing median DFS of 7 months (95% CI 4.06–9.94 months), not statistically significant (*p* = 0.527) (Figure 6B).

## 4. Discussion

This study reveals the difference in circulating values of sTGF-beta and sPD-L1 between radically resected and metastatic PDAC patients. Interestingly, metastatic PDAC patients showed lower levels of circulating sTGF-beta and higher levels of circulating PD-L1. Moreover, in radically resected PDAC patients, we observed that decrease in sTGF-beta (but not sPD-L1) was significantly associated with disease relapse. We also observed lower sTGF-beta at disease progression after first-line chemotherapy in metastatic PDAC patients, though this change was not statistically significant. Last but not least, we found a significant correlation between the baseline levels of sTGF-beta and sPD-L1 at the beginning of first-line chemotherapy, which supports the interaction of TGF-beta and PD-L1 pathways and suggests that sTGF-beta and sPD-L1 may synergize both mechanistically and as new potential blood-based biomarkers in PDAC.

The grim prognosis of PDAC patients is partially determined by the lack of disease-specific biomarkers both for radically resected and metastatic patients [6,35]. For this reason, a better understanding of these markers and their potential correlation is crucial for improvement of the current clinical management and hopefully for improvement of therapeutic strategies for PDAC patients.

Malignant tumor cells employ various methods of immune suppression to resist antitumor immunity. One these methods is the modulation of the PD-1/PD-L1 pathway which is called “immune checkpoint” [36]. Wu et al. investigated levels of circulating serum PD-L1 in PDAC patients, patients with benign pancreatic disease and healthy volunteers. The authors found significantly higher circulating PD-L1 in PDAC patients, therefore, sPD-L1 might be considered a possible diagnostic biomarker for PDAC [37]. In other studies, the expression of PD-L1 molecules in PDAC was associated with tumor proliferation, accelerated tumor cell carcinogenesis and drug resistance [38,39]. In particular, Birnbaum and collaborators analyzed the PD-L1 mRNA expression in 453 PDAC samples: PD-L1 upregulation was associated with shorter DFS and OS in multivariate analysis [37]. Of note, a recent study reported a statistically significant increase of the amount of PD-L1 expression in plasma-derived microvesicles from baseline to 3 months of treatment in n = 18 metastatic PDAC patients receiving gemcitabine and nab-paclitaxel chemotherapy [40], suggesting its potential as immunotherapy-modulating regimen [22].

All these findings on PD-L1 and sPD-L1 are in accordance with our study: overexpression of sPD-L1 seems to characterize the more aggressive pancreatic tumors, as observed in our metastatic and relapsed patients. Against this background, investigators have been eager to determine whether strategies targeting PD-1/PD-L1 are applicable in the management of PDAC [41] and, several human PD-1-antibody-drugs, such as Nivolumab and Pembrolizumab, and human anti-PD-L1-antibody-drugs (Durvalumab) have been tested in clinical trials.

However, up-to-date, the efficacy of a single PD-1/PD-L1 blockade was limited because of many of the intrinsic characteristics of PDAC, including a relatively low tumor mutation burden, a well-established desmoplastic reaction and an immunosuppressive microenvironment. It has been suggested that combination therapy strategies with surgery, chemotherapy, radiotherapy, molecular targeted therapy or other immunotherapies could overcome the resistance to anti-PD-1/PD-L1 monotherapy in PDAC and facilitate the transition of tumors from immunologically ‘‘cold’’ to ‘‘hot’’, namely from non-immunologic to immunologic [42].

Conversely our results on sTGF-beta are in contrast with a recent study showing that patients with low sTGF-beta at diagnosis had better OS and PFS, ad well that at the time of disease progression, sTGF-beta was further increased [27]. Of note, the plasma values reported in this study are much higher that the values observed in our samples, making extremely difficult a methodological comparison. However, previous in vivo studies showed that epithelial loss of TGF-beta signaling in the P48-Cre/LSL-Kras mouse model of PanIN disease is responsible for PDAC development/progression [40,43]. Other studies showed down-regulation of TGF-beta receptors as well as the effects of TGF-beta on stroma formation and angiogenesis in human colorectal cancer [44].

Further explorations are necessary to clarify whether TGF-β and PD-L1 are involved in the regulation of anti-tumor immunity in PDAC patients. This correlation has been suggested by Principe and collaborators who described that selective TGF-beta inhibition in CD8+ T-cells leads to regression of neoplastic disease, while systemic blockade of TGF-beta signaling via the drug galunisertib fails to promote cytotoxicity due to compensatory upregulation of PD-L1 on the cancer epithelium, failing to promote a substantial antitumor immune response. Interestingly, targeting both TGF-beta and PD-L1 receptors led to a reduction in the neoplastic phenotype, improving survival and reducing disease-associated morbidity in vivo [41]. Keeping with these findings, bintrafusp alfa, a first-in-class bifunctional fusion protein composed of the extracellular domain of the TGF-betaRII receptor (a TGF-beta ‘trap’) fused to a human IgG1 antibody blocking PD-L1, has shown clinical efficacy, with durable responses in patients affected by biliary cancer [45].

This study has several limitations, mainly due to (1) a relatively small number of patients and to the fact that some prognostic evaluations were based on different subgroups, which further reduced sample size and (2) the lack of an external validation cohort. In addition, no data were available about the levels of other cytokines that related to tumor immunity, such as IFN-γ, and TNF-α, as well as the complete information about the plasma levels of sTGF-beta and sPD-L1 before vs. after resection and of CEA and CA19-9 before surgery and at time of relapse. As such, efforts towards a large external validation cohort are currently being made both on the national and international level. However, this is the first study that suggests that TGF-beta and PD-L1 pathway may synergize in PDAC and might indeed be both biomarkers for PDAC aggressivity and new potential therapies. Thus, it remains to be investigated how feasible it is to combine these results with other clinical markers and novel blood-based biomarkers, creating a test which should overcome the limitations of the current invasive biopsy.

## 5. Conclusions

This study supports the hypothesis that TGF-beta and PD-L1 might play a role in PDAC immune evasion and progression, and hold prognostic value, providing a foundation to improve minimally-invasive procedures for such biomarkers, though these results need to be confirmed in larger prospective cohorts.

Importantly, if low circulating levels of TGF-beta reflect low intratumoral TGF-beta and reduced immunosuppression, and higher PD-L1 levels in the plasma correlate positively with high PD-L1 tumor expression, we might speculate that metastatic PDAC patients with lower levels of circulating sTGF-beta and higher levels of sPD-L1 could be more receptive to immunotherapy with PD-L1 inhibitors. This hypothesis on the potential predictive role of such soluble biomarkers for targeted immunotherapeutic approaches might have high clinical relevance. However, this needs to be clarified in further studies. In addition, these findings could support trials with dual TGF-beta and PD-L1 pathway inhibition which could represent an interesting novel approach in PDAC treatment.

## Figures and Tables

**Figure 1 life-12-00960-f001:**
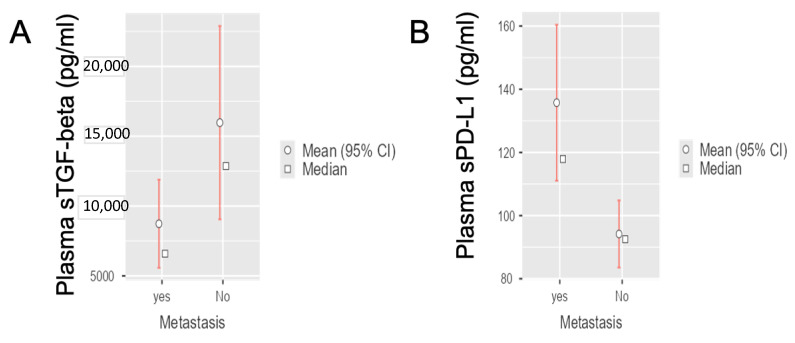
**TGF-beta and PD-L1 values differences between radically resected and metastatic PDAC patients.** (**A**) Interval plots showing statistically significant difference for sTGF-beta values between radically resected (label “metastasis No”) and metastatic (label “metastasis yes”) group of PDAC patients (Student’s *t*-test, *p* = 0.042). (**B**) Interval plots showing statistically significant difference for sPD-L1 values between radically resected (label “metastasis No”) and metastatic (label “metastasis yes”) group of PDAC patients (Student’s *t*-test, *p* = 0.010).

**Figure 2 life-12-00960-f002:**
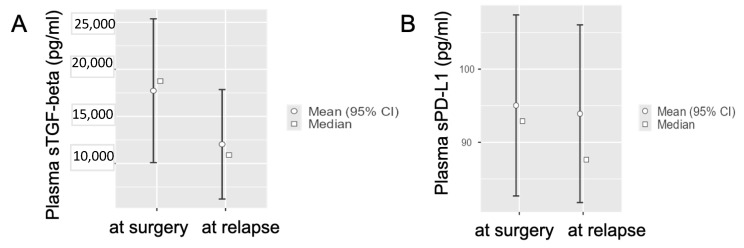
**sTGF-beta and sPD-L1 values differences at disease relapse of radically-resected PDAC patients.** (**A**) Interval plots showing statistically significant difference for sTGF-beta baseline values in radically-resected PDAC patients after surgery in comparison to TGF-beta values at disease relapse (Student’s *t*-test, *p* = 0.031). (**B**) The same comparison for PD-L1 levels did not show any statistically significant difference (Student’s *t*-test, *p* = 0.891).

**Figure 3 life-12-00960-f003:**
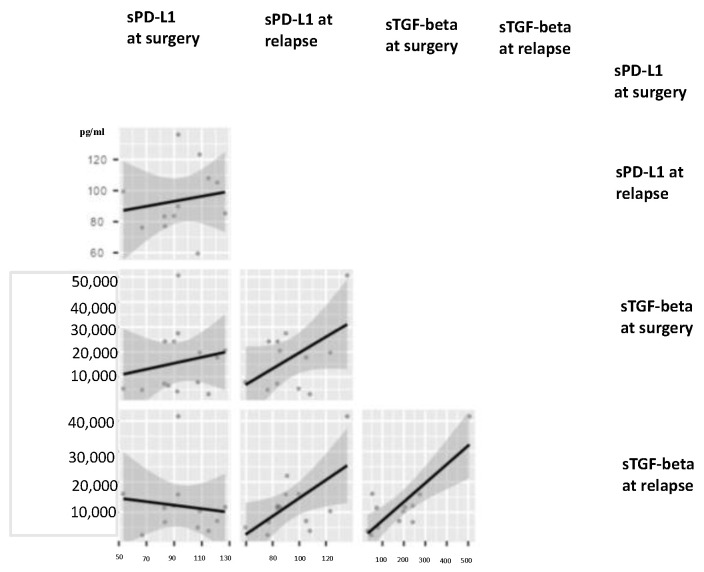
**sTGF-beta values vary at relapse of radically resected PDAC patients.** The Correlation Matrix analysis shows a statistically significant difference for sTGF-beta values in radically resected PDAC patients after surgery in comparison to sTGF-beta values at disease relapse (Correlation Matrix, *p* = 0.049).

**Figure 4 life-12-00960-f004:**
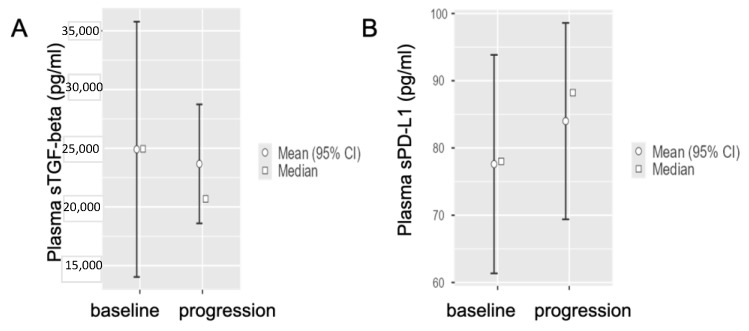
**sTGF-beta and sPD-L1 values at baseline and at the disease progression in metastatic PDAC patients.** (**A**) Interval plots showing no statistically significant difference for baseline sTGF-beta values in metastatic PDAC patients at disease progression compared to sTGF-beta baseline values at the start of first-line treatment (Student’s *t*-test, *p* = 0.799). (**B**) Interval plots showing no statistically significant difference for baseline sPD-L1 values in metastatic PDAC patients at disease progression compared to sPD-L1 baseline values at the start of first-line treatment (Student’s *t*-test, *p* = 0.891).

**Figure 5 life-12-00960-f005:**
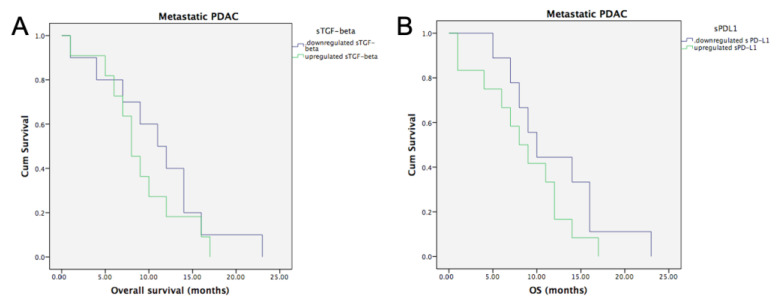
**Overall survival in metastatic PDAC according to median baseline values of sTGF-beta and sPD-L1.** (**A**) Kaplan–Meier curves showing no statistically significant difference for metastatic PDAC patients with upregulated versus downregulated baseline sTGF-beta values (Logrank test, *p* = 0.375). (**B**) Kaplan–Meier curves showing no statistically significant difference for metastatic PDAC patients with upregulated versus downregulated baseline sPD-L1 values (Logrank test, *p* = 0.218).

**Figure 6 life-12-00960-f006:**
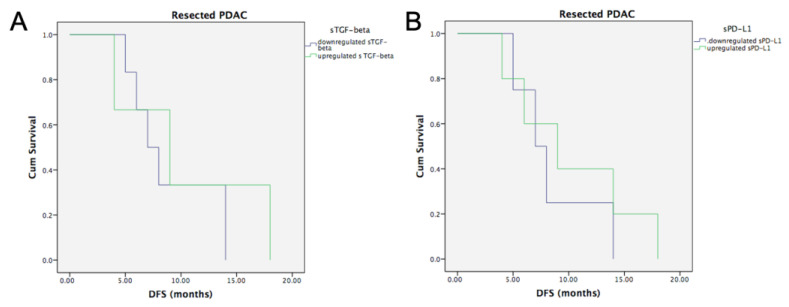
**DFS in radically resected PDAC according to median baseline values of sTGF-beta and sPD-L1.** (**A**) Kaplan–Meier curves showing no statistically significant difference for radically resected PDAC patients with upregulated versus downregulated baseline sTGF-beta values (Logrank test, *p* = 0.537). (**B**) Kaplan–Meier curves showing no statistically significant difference for radically resected PDAC with upregulated versus downregulated baseline sPD-L1 values (Logrank test, *p* = 0.527).

**Table 1 life-12-00960-t001:** Clinico-pathological features of PDAC patients.

Characteristic	Metastatic No, %	Radically Resected No, %
No. Patients	23	15
Age (median), y	69 (48–83%)	69 (43–82%)
Sex		
Male	10 (43.5%)	8 (53%)
Female	13 (56.5%)	7 (47%)
Tumor location	head: 16 (70%)	head: 10 (67%)
	body-tail: 5 (30%)	body-tail: 5 (23%)
TNM in resected PDAC pts		
T		T1N0 (1 pt), T2N0 (5 pts), unknown (1 pt)
N		T1-3N+ (8 pts)
CEA elevated after surgery		
no		11 (73%)
yes		4 (27%)
CA 19-9 elevated after surgery		
no		12 (80%)
yes		3 (20%)
Adjuvant therapy		gemcitabine-based regimen: 6 (40%)
		5-FU based regimen: 7 (47%)
		no adjuvant therapy: 2 (13%)
Relapse after surgery		
no		5 (33%)
yes		10 (67%)
Median DFS		8 (4–18 months)
CEA elevated at diagnosis of metastatic disease		
no	10 (43.5%)	
yes	13 (56.5%)	
CA 19-9 elevated at diagnosis of metastatic disease		
no	3 (13%)	
yes	20 (87%)	
Location of metastases	liver only: 8 (35%)	
	lung only: 1 (4%)	
	two or more sites: 14 (61%)	
Palliative chemotherapy	gemcitabine-based: 14 (61%)	
	5FU-based: 7 (30%)	
	no therapy: 2 (9%)

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
