# Peer review of "It Takes Two to Tango: Potential Prognostic Impact of Circulating TGF-Beta and PD-L1 in Pancreatic Cancer"

_life, 2022, doi:10.3390/life12070960_

Round 1

Reviewer 1 Report

In this study, the authors have analyzed TGF-beta and PD-L1 levels in the plasma of metastatic and radically resected PDAC patients. The idea is to understand the clinical relevance of the circulating levels of TGF-beta and PD-L1 in PDAC. The data indicate an inverse correlation between soluble TGF-beta and soluble PD-L1 levels in metastatic vs resected patients. Although there is some data to suggest the role of sTGF-beta in the prediction of PDAC relapse, the differences in patient survival and the baseline levels of sTGF-beta and sPD-L1 were not found to be significant. In my opinion, such studies are quite relevant as they provide important information about employing novel blood-based biomarkers in the diagnosis and prognosis of a lethal disease such as PDAC which has a late onset of symptoms. This study of course has limitations such as a small patient cohort, which the authors have appropriately mentioned in the discussion. The results therefore may not be conclusive to draw a clinical decision. Nevertheless, this paves the way for larger studies to corroborate the findings. A few comments to improve the manuscript are given below:

·       TGF-beta is known to be associated with immunosuppression and PDAC tumor immune evasion. Levels of TGF-beta in the circulation of metastatic PDAC patients have been found to be low as compared to radically resected patients. In addition, sPD-L1 levels are reportedly higher in metastatic patients. So, if lower circulating levels of TGF-beta reflect low TGF-beta in the tumor and higher PD-L1 levels in the plasma correlate positively with high PD-L1 tumor expression, does it mean that metastatic PDAC patients can be more receptive to immunotherapy with PD-L1 inhibitors?

·       The levels of sTGF-beta in Figures 1A, 2A, and 4A do not reconcile with the values mentioned in the respective text. Please check and ensure that the levels and the units are correct.

·       The manuscript needs to be corrected for typos.

Reviewer 2 Report

This study aims to explore the potential value of circulating TGF-β and PD-L1 in the prognosis of PDAC patients. The sTGF-β and sPD-L1 levels were firstly investigated in metastatic PDAC patients, and sTGF-β showed significantly decrease compared with radically resected patients, while sPD-L1 showed significantly increase. In addition, sTGF-β level altered in different periods after surgery. By linking the survival data of patients, this study found that sTGF-β and sPD-L1 levels were associated with the prognosis of metastatic PDAC patients. Patients with higher sTGF-β and sPD-L1 levels showed a relatively poor overall survival, but the p values were 0.357 and 0.218, respectively. On the other hand, sTGF-β and sPD-L1 levels showed no significant correlation with the survival time of patients after PDAC radically resection.

At present, there is still a lack of effective biomarkers for the prognosis of PDAC. Therefore, this study has good novelty in this field. In addition, they found that both sTGF-β and sPD-L1 display high impact on anti-tumor immune response, that will act as good biomarkers in the evaluation of PDAC immunotherapy.

The main deficiencies are as follows:

(1) The sample size included was relatively small, and the different prognostic evaluations were based on different subgroups, which further reduced sample size. 

(2) The current evidence only supports the correlation of sTGF-β and sPD-L1 levels and PDAC metastasis. The p values in Figure 2 and Figure 4 were above or close to 0.05. Figure 3 did not give the value of correlation coefficient. It is suggested to try different cut-off values and to observe whether there is an optimal cut-off to make the difference statistically significant.

(3) Both TGF-β and PD-L1 are important immunomodulatory molecules, however, this manuscript did not conduct the immunological indicator tests, such as the proportions of different immune cells (T cells, myeloid cells and so on) in peripheral blood, and the levels of cytokines that related to tumor immune (IFN-γ, TNF-α and so on). We though these data need to be supplied.

(4) The current results only suggest that TGF-β and PD-L1 levels have correlation with the metastasis and prognosis of PDAC, but the causality is still not indicated. Further explorations are necessary to clarify whether TGF-β and PD-L1 are involved in the regulation of anti-tumor immunity in PDAC patients.

Reviewer 3 Report

The study submitted for publication represents an interesting descriptive analysis of immune interplay in the surroundings of pancreatic carcinoma (PDAC) upon disease progression, therapy, and relapse reflected by two blood plasma markers. This is of interest since there is an urgent need to improve biomarkers situation by “liquid biopsy” and to shed light on the intercellular mechanisms of pathogenesis of PDAC. The manuscript is written nicely, and the study presentation is mostly sound. 

Nevertheless, there are shortcomings in reporting (and analysis?) that need to be improved:

- As your intention was a translational study, clinical information needs to be added. In general, please use a standardized reporting framework to fill information gaps (choice of sample size, potential confounders, bias control etc.) and strengthen publication of your study. Subsequently, attach the corresponding reporting list to your manuscript (see https://www.strobe-statement.org). 

- Additional surgical/ oncological data about your study cohorts are needed: location of tumor in the pancreas, TNM-status in resected PDAC (res PDAC), time of relapse in resPDAC, extent/location of metastasis in metastasized PDAC (met PDAC), details about adjuvant/ first-line chemotherapy.

- As you want to compare to bad but established biomarkers, what about plasma levels of sTGF-beta and sPD-L1 before vs. after resection in resPDAC? How about corresponding levels of CEA and CA19-9 before surgery and at time of relapse? 

- There is a contradiction between lines 236 and 297: was there a statistically significant correlation for sTGFß and sPD-L1 and the course of chemotherapy in metPDAC or just a trend?

Overall, as statistically significant differences are limited to discrimination of metPDAC vs. resPDAC for both markers and for sTGFß for relapse of resPDAC only, the conclusions should be drawn much more carefully! Maybe even good old CEA and CA 19-9 work better for this purpose?

Solely based on the data given it is invalid to state “that TGF-beta and PD-L1 play a key role in PDAC immune evasion and progression” (lines 363-364). Especially, since no correlation was seen with OS/DFS (and effect of chemotherapy?). 

And how does the data support your statement “that sTGF-beta and sPD-L1 may synergize both mechanistically” (line 297)? Please explain/ give more elaborate justification.

Round 2

Reviewer 3 Report

Thank you for your thorough revision and carefulpoint by point answers.

Some information gaps in newly provided clinical data and decrease quality of the study statements, but the manuscript is ready for publication now.